COMMUNICATIONS

# The XZZX surface code

J. Pablo Bonilla Ataides [1], David K. Tuckett [1], Stephen D. Bartlett [1], Steven T. Flammia[2] & Benjamin J. Brown [1]✉

Performing large calculations with a quantum computer will likely require a fault-tolerant architecture based on quantum error-correcting codes. The challenge is to design practical quantum error-correcting codes that perform well against realistic noise using modest resources. Here we show that a variant of the surface code—the XZZX code—offers remarkable performance for fault-tolerant quantum computation. The error threshold of this code matches what can be achieved with random codes (hashing) for every single-qubit Pauli noise channel; it is the first explicit code shown to have this universal property. We present numerical evidence that the threshold even exceeds this hashing bound for an experimentally relevant range of noise parameters. Focusing on the common situation where qubit dephasing is the dominant noise, we show that this code has a practical, high-performance decoder and surpasses all previously known thresholds in the realistic setting where syndrome measurements are unreliable. We go on to demonstrate the favourable sub-threshold resource scaling that can be obtained by specialising a code to exploit structure in the noise. We show that it is possible to maintain all of these advantages when we perform fault-tolerant quantum computation.

[1] Centre for Engineered Quantum Systems, School of Physics, University of Sydney, Sydney, NSW, Australia. [2] AWS Center for Quantum Computing, Pasadena, CA, USA. ✉email: b.brown@sydney.edu.au

A large-scale quantum computer must be able to reliably process data encoded in a nearly noiseless quantum system. To build such a quantum computer using physical qubits that experience errors from noise and faulty control, we require an architecture that operates fault-tolerantly[1–4], using quantum error correction to repair errors that occur throughout the computation.

For a fault-tolerant architecture to be practical, it will need to correct for physically relevant errors with only a modest overhead. That is, quantum error correction can be used to create near-perfect logical qubits if the rate of relevant errors on the physical qubits is below some threshold, and a good architecture should have a sufficiently high threshold for this to be achievable in practice. These fault-tolerant designs should also be efficient, using a reasonable number of physical qubits to achieve the desired logical error rate. The most common architecture for fault-tolerant quantum computing is based on the surface code[5]. It offers thresholds against depolarising noise that are already high, and encouraging recent results have shown that its performance against more structured noise can be considerably improved by tailoring the code to the noise model[6–10]. While the surface code has already demonstrated promising thresholds, its overheads are daunting[5,11]. Practical fault-tolerant quantum computing will need architectures that provide high thresholds against relevant noise models while minimising overheads through efficiencies in physical qubits and logic gates.

In this paper, we present a highly efficient fault-tolerant architecture design that exploits the common structures in the noise experienced by physical qubits. Our central tool is a variant of the surface code[12–14] where the stabilizer checks are given by the product XZZX of Pauli operators around each face on a square lattice[15]. This seemingly innocuous local change of basis offers a number of significant advantages over its more conventional counterpart for structured noise models that deviate from depolarising noise.

We first consider preserving a logical qubit in a quantum memory using the XZZX code. While some two-dimensional codes have been shown to have high error thresholds for certain types of biased noise[7,16], we find that the XZZX code gives exceptional thresholds for all single-qubit Pauli noise channels, matching what is known to be achievable with random coding according to the hashing bound[17,18]. It is particularly striking that the XZZX code can match the threshold performance of a random code, for any single-qubit Pauli error model, while retaining the practical benefits of local stabilizers and an efficient decoder. Intriguingly, for noise that is strongly biased towards X or Z, we have numerical evidence to suggest that the XZZX threshold exceeds this hashing bound, meaning this code could potentially provide a practical demonstration of the superadditivity of coherent information[19–23].

We show that these high thresholds persist with efficient, practical decoders by using a generalisation of a matching decoder in the regime where dephasing noise is dominant. In the fault-tolerant setting when stabilizer measurements are unreliable, we obtain thresholds in the biased-noise regime that surpass all previously known thresholds.

With qubits and operations that perform below the threshold error rate, the practicality of scalable quantum computation is determined by the overhead, i.e. the number of physical qubits and gates we need to obtain a target logical failure rate. Along with offering high thresholds against structured noise, we show that architectures based on the XZZX code require very low overhead to achieve a given target logical failure rate. Generically, we expect the logical failure rate to decay like $O(p^{d/2})$ at low error rates $p$ where $d = O(\sqrt{n})$ is the distance of a surface code and $n$ is the number of physical qubits used in the system. By considering a biased-noise model where dephasing errors occur a factor $\eta$ more frequently than other types of errors we demonstrate an improved logical failure rate scaling like $O((p/\sqrt{\eta})^{d/2})$. We can therefore achieve a target logical failure rate using considerably fewer qubits at large bias because its scaling is improved by a factor $\sim \eta^{-d/4}$. We also show that near-term devices, i.e. small-sized systems with error rates near to threshold, can have a logical failure rate with quadratically improved scaling as a function of distance; $O(p^{d^2/2})$. Thus, we should expect to achieve low logical failure rates using a modest number of physical qubits for experimentally plausible values of the noise bias, for example, $10 \lesssim \eta \lesssim 1000$[24,25].

Finally, we consider fault-tolerant quantum computation with biased noise[26–28], and we show that the advantages of the XZZX code persist in this context. We show how to implement low-overhead fault-tolerant Clifford gates by taking advantage of the noise structure as the XZZX code undergoes measurement-based deformations[29–31]. With an appropriate lattice orientation, noise with bias $\eta$ is shown to yield a reduction in the required number of physical qubits by a factor of $\sim \log \eta$ in a large-scale quantum computation. These advantages already manifest at code sizes attainable using present-day quantum devices.

## Results

**The XZZX surface code.** The XZZX surface code is locally equivalent to the conventional surface code[12–14], differing by a Hadamard rotation on alternate qubits[32,33]. The code parameters of the surface code are invariant under this rotation. The XZZX code therefore encodes $k = O(1)$ logical qubits using $n = O(d^2)$ physical qubits where the code distance is $d$. Constant factors in these values are determined by details such as the orientation of the square-lattice geometry and boundary conditions. See Fig. 1 for a description. This variant of the surface code was proposed in ref. [15], and has been considered as a topological memory[34]. To contrast the XZZX surface code with its conventional counterpart, we refer to the latter as the CSS surface code because it is of Calderbank-Shor-Steane type[35,36].

Together with a choice of code, we require a decoding algorithm to determine which errors have occurred and correct for them. We will consider Pauli errors $E \in \mathcal{P}$, and we say that $E$ creates a defect at face $f$ if $S_f E = (-1)ES_f$ with $S_f$ the stabilizer associated to $f$. A decoder takes as input the error syndrome (the locations of the defects) and returns a correction that will recover the encoded information with high probability. The failure probability of the decoder decays rapidly with increasing code distance, $d$, assuming the noise experienced by the physical qubits is below some threshold rate.

Because of the local change of basis, the XZZX surface code responds differently to Pauli errors compared with the CSS surface code. We can take advantage of this difference to design better decoding algorithms. Let us consider the effect of different types of Pauli errors, starting with Pauli-Z errors. A single Pauli-Z error gives rise to two nearby defects. In fact, we can regard a Pauli-Z error as a segment of a string where defects lie at the endpoints of the string segment, and where multiple Pauli-Z errors compound into longer strings, see Fig. 1d.

A key feature of the XZZX code that we will exploit is that Pauli-Z error strings align along the same direction, as shown in Fig. 1d. We can understand this phenomenon in more formal terms from the perspective of symmetries[10,37]. Indeed, the product of face operators along a diagonal such as that shown in Fig. 1e commute with Pauli-Z errors. This symmetry guarantees that defects created by Pauli-Z errors will respect a parity conservation law on the faces of a diagonal oriented along this direction. Using this property, we can decode Pauli-Z errors

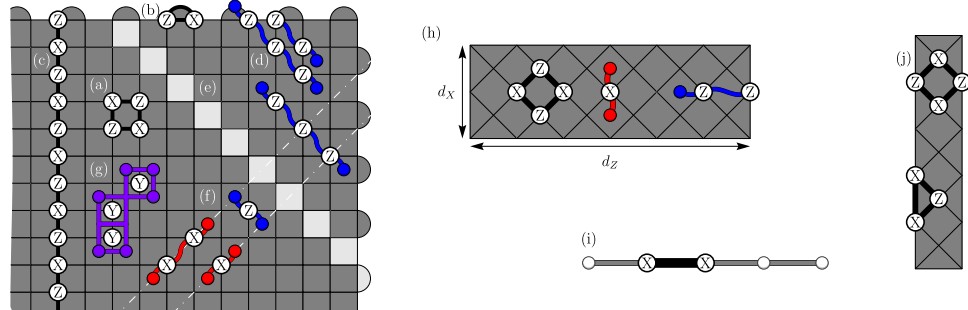

**Fig. 1 The XZZX surface code.** Qubits lie on the vertices of the square lattice. The codespace is the common +1 eigenspace of its stabilizers $S_f$ for all faces of the lattice $f$. **a** An example of a stabilizer $S_f$ associated with face $f$. We name the XZZX code according to its stabilizer operators that are the product of two Pauli-X terms and two Pauli-Z terms. Unlike the conventional surface code, the stabilizers are the same at every face. **b** A boundary stabilizer. **c** A logical operator that terminates at the boundary. **d** Pauli-Z errors give rise to string-like errors that align along a common direction, enabling a one-dimensional decoding strategy. **e** The product of stabilizer operators along a diagonal give rise to symmetries under an infinite bias dephasing noise model[10,37]. **f** Pauli-X errors align along lines with an orthogonal orientation. At finite bias, errors in conjugate bases couple the lines. **g** Pauli-Y errors can be decoded as in ref. [10]. **h** A convenient choice of boundary conditions for the XZZX code are rectangular on a rotated lattice geometry. Changing the orientation of the lattice geometry means high-rate Pauli-Z errors only create strings oriented horizontally along the lattice. We can make a practical choice of lattice dimensions with $d_Z > d_X$ to optimise the rates of logical failure caused by either low- or high-rate errors. Small-scale implementations of the XZZX code on rectangular lattices may be well suited for implementation with near-term devices. **i** In the limit where $d_X = 1$ we find a repetition code. This may be a practical choice of code given a limited number of qubits that experience biased noise. **j** The next engineering challenge beyond a repetition code is an XZZX code on a rectangle with $d_X = 2$. This code can detect a single low-rate error.

on the XZZX code as a series of disjoint repetition codes. It follows that, for a noise model described by independent Pauli-Z errors, this code has a threshold error rate of 50%.

Likewise, Pauli-X errors act similarly to Pauli-Z errors, but with Pauli-X error strings aligned along the orthogonal direction to the Pauli-Z error strings. In general, we would like to be able to decode all local Pauli errors, where error configurations of Pauli-X and Pauli-Z errors violate the one-dimensional symmetries we have introduced, e.g. Fig. 1f. As we will see, we can generalise conventional decoding methods to account for finite but high bias of one Pauli operator relative to others and maintain a very high threshold.

We finally remark that the XZZX surface code responds to Pauli-Y errors in the same way as the CSS surface code. Each Pauli-Y error will create four defects on each of their adjacent faces; see Fig. 1g. The high-performance decoders presented in refs. [7,8,10] are therefore readily adapted for the XZZX code for an error model where Pauli-Y errors dominate.

**Optimal thresholds**. The XZZX code has exceptional thresholds for all single-qubit Pauli noise channels. We demonstrate this fact using an efficient maximum-likelihood decoder[38], which gives the optimal threshold attainable with the code for a given noise model. Remarkably, we find that the XZZX surface code achieves code-capacity threshold error rates that closely match the zero-rate hashing bound for all single-qubit Pauli noise channels, and appears to exceed this bound in some regimes.

We define the general single-qubit Pauli noise channel

$$\mathcal{E}(\rho) = (1-p)\rho + p(r_X X\rho X + r_Y Y\rho Y + r_Z Z\rho Z) \qquad (1)$$

where $p$ is the probability of any error on a single qubit and the channel is parameterised by the stochastic vector $\boldsymbol{r} = (r_X, r_Y, r_Z)$, where $r_X, r_Y, r_Z \geq 0$ and $r_X + r_Y + r_Z = 1$. The surface of all possible values of $\boldsymbol{r}$ parametrise an equilateral triangle, where the centre point $(1/3, 1/3, 1/3)$ corresponds to standard depolarising noise, and vertices $(1, 0, 0)$, $(0, 1, 0)$ and $(0, 0, 1)$ correspond to pure $X$, $Y$ and $Z$ noise, respectively. We also define biased-noise channels, which are restrictions of this general noise channel, parameterised by the scalar $\eta$; for example, in the case of

Z-biased noise, we define $\eta = r_Z/(r_X + r_Y)$ where $r_X = r_Y$, such that $\eta = 1/2$ corresponds to standard depolarising noise and the limit $\eta \to \infty$ corresponds to pure $Z$ noise. The hashing bound is defined as $R = 1 - H(\boldsymbol{p})$ with $R$ an achievable rate, $k/n$, using random codes and $H(\boldsymbol{p})$ the Shannon entropy for the vector $\boldsymbol{p} = p\boldsymbol{r}$. For our noise model, for any $\boldsymbol{r}$ there is a noise strength $p$ for which the achievable rate via random coding goes to zero; we refer to this as the zero-rate hashing bound, and it serves as a useful benchmark for code-capacity thresholds.

We estimate the threshold error rate as a function of $\boldsymbol{r}$ for both the XZZX surface code and the CSS surface code using a tensor-network decoder that gives a controlled approximation to the maximum-likelihood decoder[7,8,38]; see Methods for details. Our results are summarised in Fig. 2. We find that the thresholds of the XZZX surface code closely match or slightly exceed (as discussed below), the zero-rate hashing bound for all investigated values of $\boldsymbol{r}$, with a global minimum $p_c = 18.7(1)\%$ at standard depolarising noise and peaks $p_c \sim 50\%$ at pure $X$, $Y$ and $Z$ noise. We find that the thresholds of the CSS surface code closely match this hashing bound for $Y$-biased noise, where $Y$ errors dominate, consistent with prior work[7,8], as well as for channels where $r_Y < r_X = r_Z$ such that $X$ and $Z$ errors dominate but are balanced. In contrast to the XZZX surface code, we find that the thresholds of the CSS surface code fall well below this hashing bound as either $X$ or $Z$ errors dominate with a global minimum $p_c = 10.8(1)\%$ at pure $X$ and pure $Z$ noise.

In some cases, our estimates of XZZX surface code thresholds appear to exceed the zero-rate hashing bound. The discovery of such a code would imply that we can create a superadditive coherent channel via code concatenation. To see why, consider an inner code with a high threshold that exceeds the hashing bound, $p_c > p_{h.b.}$, together with a finite-rate outer code with rate $R_{out} = K_{out}/N_{out} > 0$ that has some arbitrary nonzero threshold against independent noise[39–42]. Now consider physical qubits with an error rate $p$ below the threshold of the inner code but above the hashing bound, i.e. $p_{h.b.} < p < p_c$. We choose a constant-sized inner code using $N_{in}$ qubits such that its logical failure rate is below the threshold of the outer code. Concatenating this inner code into the finite-rate outer code will give us a family of codes with rate

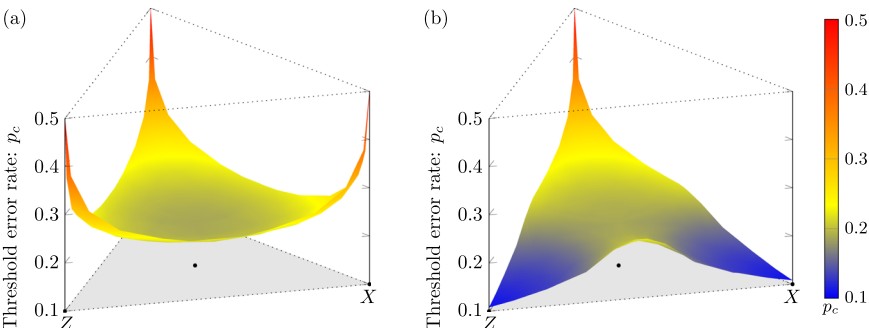

**Fig. 2 Optimal code-capacity thresholds over all single-qubit Pauli channels.** Threshold estimates $p_c$ are found using approximate maximum-likelihood decoding for **a** the XZZX surface code and **b** the CSS surface code with open boundaries (as in Fig. 1b). The grey triangle represents a parametrisation of all single-qubit Pauli channels, where the centre corresponds to depolarising noise, the labeled vertices correspond to pure $X$ and $Z$ noise, and the third vertex corresponds to pure $Y$ noise. For the XZZX code, estimates closely match the zero-rate hashing bound (not shown) for all single-qubit Pauli channels. For the CSS code, estimates closely match the hashing bound for $Y$-biased noise but fall well below for $X$- and $Z$-biased noise. All estimates use $d \times d$ codes with distances $d \in \{13, 17, 21, 25\}$.

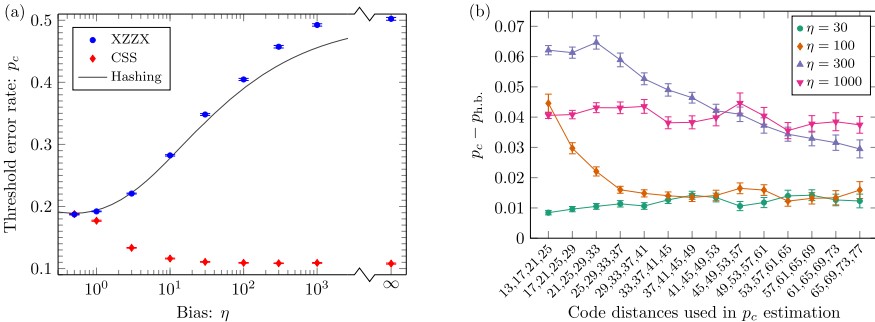

**Fig. 3 Estimates of optimal XZZX surface code thresholds relative to the hashing bound. a** Threshold estimates $p_c$ for the XZZX and CSS surface codes as a function of bias $\eta$ with $Z$-biased (or, by code symmetry, $X$-biased) noise using approximate maximum-likelihood decoding and codes with open boundaries (as in Fig. 1b). The solid line is the zero-rate hashing bound for the associated Pauli noise channel, where the entropy of the channel equals 1 bit. For high bias, $\eta \geq 30$, the estimates for the XZZX code exceed the hashing bound. To investigate this surprising effect, estimates for the XZZX code with $30 \leq \eta \leq 1000$ use large $d \times d$ codes with distances $d \in \{65, 69, 73, 77\}$; other estimates use distances $d \in \{13, 17, 21, 25\}$ (as used for Fig. 2). **b** Difference between threshold estimates for the XZZX code with $Z$-biased noise and the hashing bound $p_c - p_{h.b.}$ as a function of code distances used in the estimation. Data is shown for biases $\eta = 30, 100, 300, 1000$. Threshold estimates exceed the hashing bound in all cases. The gap reduces, in most cases, with sets of greater code distance, but it persists and appears to stabilise for $\eta = 30$, 100 and 1000. In both plots, error bars indicate one standard deviation relative to the fitting procedure.

$R' = R_{out}/N_{in} > 0$ and a vanishing failure probability as $N_{out} \to \infty$. If both codes have low-density parity checks (LDPCs)[41,42], the resulting code provides an example of a superadditive LDPC code.

Given the implications of a code that exceeds the zero-rate hashing bound we now investigate our numerics in this regime further. For the values of $r$ investigated for Fig. 2, the mean difference between our estimates and the hashing bound is $\overline{p_c - p_{h.b.}} = -0.1(3)\%$ and our estimates never fall more than 1.1% below the hashing bound. However, for high bias, $\eta \geq 100$, we observe an asymmetry between $Y$-biased noise and $Z$-biased (or, equivalently, $X$-biased) noise. In particular, we observe that, while threshold estimates with $Y$-biased noise match the hashing bound to within error bars, threshold estimates with highly biased $Z$ noise significantly exceed the hashing bound. Our results with $Z$-biased noise are summarised in Fig. 3, where, since thresholds are defined in the limit of infinite code distance, we provide estimates with sets of increasing code distance for $\eta \geq 30$. Although the gap typically reduces, it appears to stabilise for $\eta = 30$, 100, 1000, where we find $p_c - p_{h.b.} = 1.2(2)\%$, 1.6(3)%, 3.7(3)%, respectively, with the largest code distances; for $\eta = 300$, the gap exceeds 2.9% but has clearly not yet stabilised. This evidence for exceeding the zero-rate hashing bound appears to be robust, but warrants further study.

Finally, we evaluate threshold error rates for the XZZX code with rectangular boundaries using a minimum-weight perfect-matching decoder, see Fig. 4. Matching decoders are very fast, and so allow us to explore very large systems sizes; they are also readily generalised to the fault-tolerant setting as discussed below. Our decoder is described in Methods. Remarkably, the thresholds we obtain closely follow the zero-rate hashing bound at high bias. This is despite using a sub-optimal decoder that does not use all of the syndrome information. Again, our data appear to marginally exceed this bound at high bias.

**Fault-tolerant thresholds.** Having demonstrated the remarkable code-capacity thresholds of the XZZX surface code, we now demonstrate how to translate these high thresholds into practice using a matching decoder[14,43,44]. We find exceptionally high fault-tolerant thresholds, i.e. allowing for noisy measurements, with respect to a biased phenomenological noise model. Moreover, for unbiased noise models we recover the standard matching decoder[14,45].

To detect measurement errors we repeat measurements over a long time[14]. We can interpret measurement errors as strings that align along the temporal axis with a defect at each endpoint. This

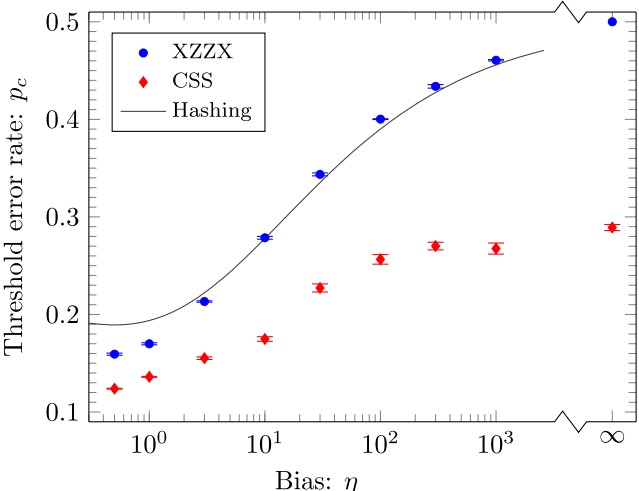

**Fig. 4 Thresholds for the XZZX code using a matching decoder.** Code-capacity thresholds $p_c$ for the XZZX code with rectangular boundary conditions (as in Fig. 1h) shown as a function of noise bias $\eta$ using a matching decoder. The threshold error rates for the XZZX code experiencing Pauli-Z-biased noise (blue) significantly outperform those found using the matching decoder presented in ref. [10] experiencing Pauli-Y-biased noise for the CSS surface code (red). For the XZZX code, we evaluated separate thresholds for logical Pauli-X and Pauli-Z errors, with the lowest of the two shown here (although the discrepancy between the two different thresholds is negligible). Data points are found with ~$10^5$ Monte-Carlo samples for each physical error rate sampled and each lattice size used. We study the XZZX code for large lattices with $d_Z = A_\eta d_X$ where aspect ratios take values $1 \leq A_\eta \leq 157$ such that $A_{1/2} = 1$ and $A_{1000} = 157$. We find the XZZX code matches the zero-rate hashing bound at $\eta \sim 10$ (solid line). For larger biases the data appear to exceed the hashing bound. For instance, at $\eta = 100$ we found $p_c - p_{\text{h.b.}} \sim 1\%$. We obtained this threshold using code sizes $d_X = 7, 11, 15$ and $A_{100} = 23$. Error bars indicate one standard deviation obtained by jackknife resampling over code distance.

allows us to adapt minimum-weight perfect-matching for fault-tolerant decoding. We explain our simulation in Fig. 5a–d and describe our decoder in Methods.

We evaluate fault-tolerant thresholds by finding logical failure rates using Monte-Carlo sampling for different system parameters. We simulate the XZZX code on a $d \times d$ lattice with periodic boundary conditions, and we perform $d$ rounds of stabilizer measurements. We regard a given sample as a failure if the decoder introduces a logical error to the code qubits, or if the combination of the error string and its correction returned by the decoder includes a non-trivial cycle along the temporal axis. It is important to check for temporal errors, as they can cause logical errors when we perform fault-tolerant logic gates by code deformation[46].

The phenomenological noise model is defined such that qubits experience errors with probability $p$ per unit time. These errors may be either high-rate Pauli-Z errors that occur with probability $p_{\text{h.r.}}$ per unit time, or low-rate Pauli-X or Pauli-Y errors each occurring with probability $p_{\text{l.r.}}$ per unit time. The noise bias with this phenomenological noise model is defined as $\eta = p_{\text{h.r.}}/(2p_{\text{l.r.}})$. One time unit is the time it takes to make a stabilizer measurement, and we assume we can measure all the stabilizers in parallel[5]. Each stabilizer measurement returns the incorrect outcome with probability $q = p_{\text{h.r.}} + p_{\text{l.r.}}$. To leading order, this measurement error rate is consistent with a measurement circuit where an ancilla is prepared in the state $|+\rangle$ and subsequently entangled to the qubits of $S_f$ with bias-preserving controlled-not and controlled-phase gates before its measurement in the Pauli-X

basis. With such a circuit, Pauli-Y and Pauli-Z errors on the ancilla will alter the measurement outcome. At $\eta = 1/2$ this noise model interpolates to a conventional noise model where $q = 2p/3$[47]. We also remark that hook errors[47,48], i.e. correlated errors that are introduced by this readout circuit, are low-rate events. This is because high-rate Pauli-Z errors acting on the control qubit commute with the entangling gate, and so no high-rate errors are spread to the code.

Intuitively, the decoder will preferentially pair defects along the diagonals associated with the dominant error. In the limit of infinite bias at $q = 0$, the decoder corrects the Pauli-Z errors by treating the XZZX code as independent repetition codes. It follows that by extending the syndrome along the temporal direction to account for the phenomenological noise model with infinite bias, we effectively decode $d$ decoupled copies of the two-dimensional surface code, see Fig. 5. With the minimum-weight perfect-matching decoder, we therefore expect a fault-tolerant threshold ~10.3%[14]. Moreover, when $\eta = 1/2$ the minimum-weight perfect-matching decoder is equivalent to the conventional matching decoder[14,45]. We use these observations to check that our decoder behaves correctly in these limits.

In Fig. 5e, we present the thresholds we obtain for the phenomenological noise model as a function of the noise bias $\eta$. In the fault-tolerant case, we find our decoder tends towards a threshold of ~10% as the bias becomes large. We note that the threshold error rate appears lower than the expected ~10.3%; we suggest that this is a small-size effect. Indeed, the success of the decoder depends on effectively decoding ~$d$ independent copies of the surface code correctly. In practice, this leads us to underestimate the threshold when we perform simulations using finite-sized systems.

Notably, our decoder significantly surpasses the thresholds found for the CSS surface code against biased Pauli-Y errors[10]. We also compare our results to a conventional minimum-weight perfect-matching decoder for the CSS surface code where we correct bit-flip errors and dephasing errors separately. As we see, our decoder for the XZZX code is equivalent to the conventional decoding strategy at $\eta = 1/2$ and outperforms it for all other values of noise bias.

**Overheads**. We now show that the exceptional error thresholds of the XZZX surface code are accompanied by significant advantages in terms of the scaling of the logical failure rate as a function of the number of physical qubits $n$ when error rates are below threshold. Improvements in scaling will reduce the resource overhead, because fewer physical qubits will be needed to achieve a desired logical failure rate.

The XZZX code with periodic boundary conditions on a lattice with dimensions $d \times (d + 1)$ has the remarkable property that it possesses only a single logical operator that consists of only physical Pauli-Z terms. Moreover, this operator has weight $n = d(d + 1)$. Based on the results of ref. [8], we can expect that the XZZX code on such a lattice will have a logical failure rate that decays like $O(p_{\text{h.r.}}^{d^2/2})$ at infinite bias. Note we can regard this single logical-Z operator as a string that coils around the torus many times such that it is supported on all $n$ qubits. As such, this model can be regarded as an $n$-qubit repetition code whose logical failure rate decays like $O(p^{n/2})$.

Here we use the XZZX code on a periodic $d \times (d + 1)$ lattice to test the performance of codes with high-weight Pauli-Z operators at finite bias. We find, at high bias and error rates near to threshold, that a small XZZX code can demonstrate this rapid decay in logical failure rate. In general, at more modest biases and at lower error rates, we find that the logical failure rate scales like $O((p/\sqrt{\eta})^{d/2})$ as the system size diverges. This scaling indicates a significant advantage in the overhead cost of architectures that

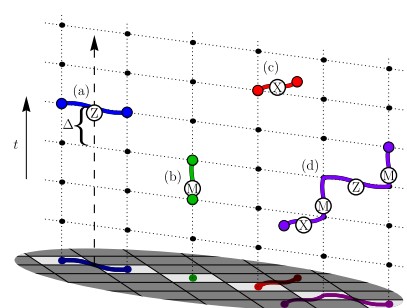

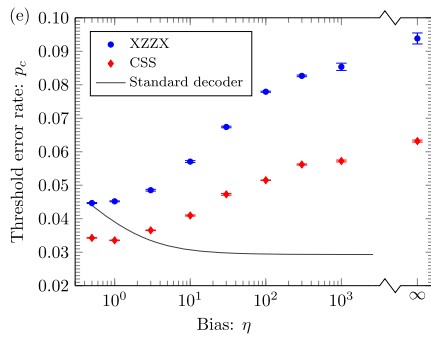

**Fig. 5 Fault-tolerant thresholds for the XZZX code. a–d** Spacetime where stabilizer measurements are unreliable. Time $t$ progresses upwards and stabilizers are measured at moments marked by black vertices. We identify a defect when a stabilizer measurement differs from its previous outcome. **a** The dashed line shows the worldline of one qubit. If a Pauli-Z error occurs in the time interval $\Delta$, a horizontal string is created in the spacetime with defects at its endpoints at the following round of stabilizer measurements. **b** Measurement errors produce two sequential defects that we interpret as strings that align along the vertical direction. **c** Pauli-X errors create string-like errors that align orthogonally to the Pauli-Z errors and measurement errors. **d** In general errors compound to make longer strings. In the limit where there are no Pauli-X errors all strings are confined to the square lattice we show. **e** Fault-tolerant threshold error rates $p_c$ as a function of noise bias $\eta$ and measurement error rates $q = p_{\text{h.r.}} + p_{\text{l.r.}}$. The results found using our matching decoder for the XZZX code experiencing Pauli-Z-biased noise (blue) are compared with the results found using the matching decoder presented in ref. [10] experiencing Pauli-Y-biased noise for the CSS surface code (red). Equivalent results to the red points are obtained with Pauli-Z-biased noise using the tailored code of ref. [7]. The XZZX code significantly outperforms the CSS code for all noise biases. At a fixed bias, data points are found with $3 \times 10^4$ Monte-Carlo samples for each physical error rate sampled and for each square lattice with distance $d \in \{12, 14, ..., 20\}$ at finite bias and $d \in \{24, 28, ..., 40\}$ at infinite bias. Error bars indicate one standard deviation obtained by jackknife resampling over code distance. The solid line shows the threshold of the conventional matching decoder for the CSS surface code undergoing phenomenological noise where bit-flip and dephasing errors are decoded independently. Specifically, it follows the function $p_{\text{h.r.}} + p_{\text{l.r.}} = 0.029$ where ~2.9% is the phenomenological threshold[45]. We note that our decoder is equivalent to the conventional matching decoder at $\eta = 1/2$.

take advantage of biased noise. We demonstrate both of these regimes with numerical data.

In practice, it will be advantageous to find low-overhead scaling using codes with open boundary conditions. We finally argue that the XZZX code with rectangular open boundary conditions will achieve comparable overhead scaling in the large system size limit.

Let us examine the different failure mechanisms for the XZZX code on the periodic $d \times (d+1)$ lattice more carefully. Restricting to Pauli-Z errors, the weight of the only non-trivial logical operator is $d(d+1)$. This means the code can tolerate up to $d(d+1)/2$ dephasing errors, and we can therefore expect failures due to high-rate errors to occur with probability

$$\overline{P}_{\text{quad.}} \sim N_{\text{h.r.}} p_{\text{h.r.}}^{d^2/2}, \tag{2}$$

below threshold, where $N_{\text{h.r.}} \sim 2^{d^2}$ is the number of configurations that $d^2/2$ Pauli-Z errors can take on the support of the weight-$d^2$ logical operator to cause a failure. We compare this failure rate to the probability of a logical error caused by a string of $d/4$ high-rate errors and $d/4$ low-rate errors. We thus consider the ansatz

$$\overline{P}_{\text{lin.}} \sim N_{\text{l.r.}} (1-p)^{d^2-d/2} (p_{\text{h.r.}} + p_{\text{l.r.}})^{d/4} (2p_{\text{l.r.}})^{d/4} \tag{3}$$

where $N_{\text{l.r.}} \sim 2^{\gamma d}$ is an entropy term with $3/2 \lesssim \gamma \lesssim 2$[49]. We justify this ansatz and estimate $\gamma$ in Methods.

This structured noise model thus leads to two distinct regimes, depending on which failure process is dominant. In the first regime where $\overline{P}_{\text{quad.}} \gg \overline{P}_{\text{lin.}}$, we expect that the logical failure rate will decay like $\sim p_{\text{h.r.}}^{d^2/2}$. We find this behaviour with systems of a finite size and at high bias where error rates are near to threshold. We evaluate logical failure rates using numerical simulations to demonstrate the behavior that characterises this regime; see Fig. 6 (a). Our data show good agreement with the scaling ansatz $\overline{P} = Ae^{Bd^2}$. In contrast, our data are not well described by a scaling $\overline{P} = Ae^{Bd}$.

We observe the regime where $\overline{P}_{\text{lin.}} \gg \overline{P}_{\text{quad.}}$ using numerics at small $p$ and modest $\eta$. In this regime, logical errors are caused by a mixture of low-rate and high-rate errors that align along a path of weight $O(d)$ on some non-trivial cycle. In Fig. 6b, we show that the data agree well with the ansatz of Eq. (3), with $\gamma \sim 1.8$. This remarkable correspondence to our data shows that our decoder is capable of decoding up to ~$d/4$ low-rate errors, even with a relatively large number of high-rate errors occurring simultaneously on the lattice.

In summary, for either scaling regime, we find that there are significant implications for overheads. We emphasise that the generic case for fault-tolerant quantum computing is expected to be the regime dominated by $\overline{P}_{\text{lin.}}$. In this regime, the logical failure rate of a code is expected to decay as $\overline{P} \sim p^{d/2}$ below threshold[5,50,51]. Under biased noise, our numerics show that failure rates $\overline{P} \sim (p/\sqrt{\eta})^{d/2}$ can be obtained. This additional decay factor ~$\eta^{-d/4}$ in our expression for logical failure rate means we can achieve a target logical failure rate with far fewer qubits at high bias.

The regime dominated by $\overline{P}_{\text{quad.}}$ scaling is particularly relevant for near-term devices that have a small number of qubits operating near the threshold error rate. In this situation, we have demonstrated a very rapid decay in logical failure rate like $\sim p^{d^2/2}$ at high bias, if they can tolerate ~$d^2/2$ dephasing errors.

We finally show that we can obtain a low-overhead implementation of the XZZX surface code with open boundary conditions using an appropriate choice of lattice geometry. As we explain below, this is important for performing fault-tolerant quantum computation with a two-dimensional architecture. Specifically, with the geometry shown in Fig. 1h, we can reduce the length of one side of the lattice by a factor of $O(1/\log \eta)$, leaving a smaller rectangular array of qubits. This is because high-rate error strings of the biased-noise model align along the horizontal direction only. We note that $d_X$ ($d_Z$) denote the least weight logical operator comprised of only Pauli-X (Pauli-Z) operators. We can therefore choose $d_X \ll d_Z$ without

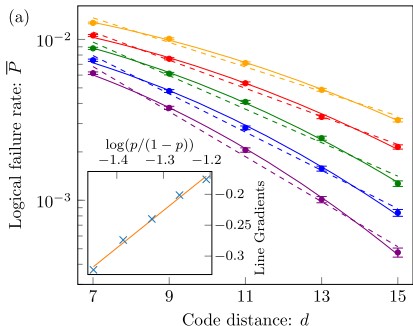
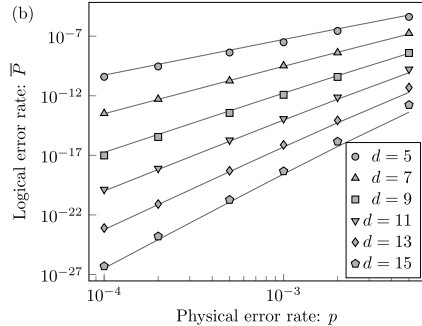

**Fig. 6 Sub-threshold scaling of the logical failure rate with the XZZX code. a** Logical failure rate $\bar{P}$ at high bias near to threshold plotted as a function of code distance $d$. We use a lattice with coprime dimensions $d \times (d+1)$ for $d \in \{7, 9, 11, 13, 15\}$ at bias $\eta = 300$, assuming ideal measurements. The data were collected using $N = 5 \times 10^5$ iterations of Monte-Carlo (MC) samples for each physical rate sampled and for each lattice dimension used. The physical error rates used are, from the bottom to the top curves in the main plot, $p = 0.19, 0.20, 0.21, 0.22$ and $0.23$. Error bars represent one standard deviation for the Monte-Carlo simulations. The solid lines are a fit of the data to $\bar{P}_{\text{quad.}} = A e^{Bd^2}$, consistent with Eq. (2), and the dashed lines a fit to $\bar{P}_{\text{lin.}} = A e^{Bd}$, consistent with Eq. (3) where we would expect $B = \log(p/(1-p))/2$, see Methods. The data fit the former very well; for the latter, the gradients of the best fit dashed lines, as shown on the inset plot as a function of $\log(p/(1-p))$, give a linear slope of 0.61(3). Because this slope exceeds the value of 0.5, we conclude that the sub-threshold scaling is not consistent with $\bar{P}_{\text{lin.}} = A e^{Bd}$. **b** Logical failure rates $\bar{P}$ at modest bias far below threshold plotted as a function of the physical error rate $p$. The data (markers) were collected at bias $\eta = 3$ and coprime $d \times (d+1)$ code dimensions of $d \in \{5, 7, 9, 11, 13, 15\}$ assuming ideal measurements. Data is collected using the Metropolis algorithm and splitting method presented in refs. [76,77]. The solid lines represent the prediction of Eq. (3). The data show very good agreement with the single parameter fitting for all system sizes as $p$ tends to zero.

compromising the logical failure rate of the code due to Pauli-$Z$ errors at high bias. This choice may have a dramatic effect on the resource cost of large-scale quantum computation. We estimate that the optimal choice is

$$d_Z \approx d_X \left(1 - \frac{\log \eta}{\log p}\right). \quad (4)$$

using approximations that apply at low error rates. To see this, let us suppose that a logical failure due to high(low)-rate errors is $\bar{P}_{\text{h.r.}} \approx p^{d_Z/2}$ ($\bar{P}_{\text{l.r.}} \approx (p/\eta)^{d_X/2}$) where we have neglected entropy terms and assumed $p_{\text{h.r.}} \sim p$ and $p_{\text{l.r.}} \sim p/\eta$. Equating $\bar{P}_{\text{l.r.}}$ and $\bar{P}_{\text{h.r.}}$ gives us Eq. (4). Similar results have been obtained in, e.g. refs. [16,26,52–54] with other codes. Assuming an error rate that is far below threshold, e.g. $p \sim 1\%$, and a reasonable bias we might expect $\eta \sim 100$, we find an aspect ratio $d_X \sim d_Z/2$.

**Low-overhead fault-tolerant quantum computation.** As with the CSS surface code, we can perform fault-tolerant quantum computation with the XZZX code using code deformations[29–31,55–57]. Here we show how to maintain the advantages that the XZZX code demonstrates as a memory experiencing structured noise, namely, its high-threshold error rates and its reduced resource costs, while performing fault-tolerant logic gates.

A code deformation is a type of fault-tolerant logic gate where we manipulate encoded information by changing the stabilizer group we measure[55,57]. These altered stabilizer measurements project the system onto another stabilizer code where the encoded information has been transformed or 'deformed'. These deformations allow for Clifford operations with the surface code; Clifford gates are universal for quantum computation when supplemented with the noisy initialisation of magic states[58]. Although initialisation circuits have been proposed to exploit a bias in the noise[59], here we focus on fault-tolerant Clifford operations and the fault-tolerant preparation of logical qubits in the computational basis.

Many approaches for code deformations have been proposed that, in principle, could be implemented in a way to take advantage of structured noise using a tailored surface code. These approaches include braiding punctures[55–57,60], lattice

surgery[29,30,61,62] and computation with twist defects[30,63,64]. We focus on a single example based on lattice surgery as in refs. [31,62]; see Fig. 7a. We will provide a high-level overview and leave open all detailed questions of implementation and threshold estimates for fault-tolerant quantum computation to future work.

Our layout for fault-tolerant quantum computation requires the fault-tolerant initialisation of a hexon surface code, i.e. a surface code with six twist defects at its boundaries[30]; see Fig. 7 (b). We can fault-tolerantly initialise this code in eigenstates of the computational basis through a process detailed in Fig. 7. We remark that the reverse operation, where we measure qubits of the XZZX surface code in this same product basis, will read the code out while respecting the properties required to be robust to the noise bias. Using the arguments presented above for the XZZX code with rectangular boundaries, we find a low-overhead implementation with dimensions related as $d_Z = A_\eta d_X$, where we might choose an aspect ratio $A_\eta = O(\log \eta)$ at low error rates and high noise bias.

We briefly confirm that this method of initialisation is robust to our biased-noise model. Principally, this method must correct high-rate Pauli-$Z$ errors on the red qubits, as Pauli-$Z$ errors act trivially on the blue qubits in eigenstates of the Pauli-$Z$ operator during preparation. Given that the initial state is already in an eigenstate of some of the stabilizers of the XZZX surface code, we can detect these Pauli-$Z$ errors on red qubits, see, e.g. Fig. 7(v). The shaded faces will identify defects due to the Pauli-$Z$ errors. Moreover, as we discussed before, strings created by Pauli-$Z$ errors align along horizontal lines using the XZZX surface code. This, again, is due to the stabilizers of the initial state respecting the one-dimensional symmetries of the code under pure dephasing noise. In addition to robustness against high-rate errors, low-rate errors as in Fig. 7(vi) can also be detected on blue qubits. The bit-flip errors violate the stabilizers we initialise when we prepare the initial product state. As such we can adapt the high-threshold error-correction schemes we have proposed for initialisation to detect these errors for the case of finite bias. We therefore benefit from the advantages of the XZZX surface code under a biased error model during initialisation.

Code deformations amount to initialising and reading out different patches of a large surface code lattice. As such, performing arbitrary code deformations while preserving the

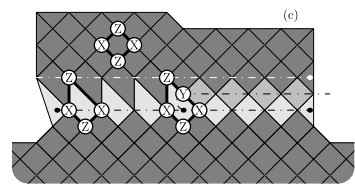

**Fig. 7 Generalised lattice surgery.** Details of generalised lattice surgery are given in refs. [31,62]. **a** Pairs of qubits are encoded on surface codes with six twist defects lying on their boundaries[30] (i). Entangling operations are performed by making parity measurements with an ancillary surface code, (ii). Circled areas are described in terms of the microscopic details of the architecture in parts **b** and **c** of the figure, respectively. **b** Initialising a hexon surface code. Red (blue) vertices are initialised in Pauli-X(Pauli-Z) basis. The system is prepared in an eigenstate of the stabilizers shown on the shaded faces, (iii) and the logical Pauli-Z operators, (iv). This initialisation strategy is robust to biased noise. Pauli-Z errors that can occur on red vertices are detected by the shaded faces (v). We can also detect low-rate Pauli-X errors on blue vertices with this method of initialisation (vi). We can decode all of these initialisation errors on this subset of faces using the minimum-weight perfect-matching decoder in the same way we decode the XZZX code as a memory. **c** The hexon surface code fused to the ancillary surface code to perform a logical Pauli-Y measurement. The lattice surgery procedure introduces a twist in the centre of the lattice. We show the symmetry with respect to the Pauli-Z errors by lightly colored faces. Again, decoding this model in the infinite bias limit is reduced to decoding one-dimensional repetition codes, except at the twist where there is a single branching point.

biased-noise protection offered by the XZZX surface code is no more complicated than what has already been demonstrated. This is with one exception. We might consider generalisations of lattice surgery or other code deformations where we can perform fault-tolerant Pauli-Y measurements. In this case, we introduce a twist to the lattice[63] and, as such, we need to reexamine the symmetries of the system to propose a high-performance decoder. We show the twist in the centre of Fig. 7c together with its weight-five stabilizer operator. A twist introduces a branch in the one-dimensional symmetries of the XZZX surface code. A minimum-weight perfect-matching decoder can easily be adapted to account for this branch. Moreover, should we consider performing fault-tolerant Pauli-Y measurements, we do not expect that a branch on a single location on the lattice will have a significant impact on the performance of the code experiencing structured noise. Indeed, even with a twist on the lattice, the majority of the lattice is decoded as a series of one-dimensional repetition codes in the infinite bias limit.

## Discussion

We have shown how fault-tolerant quantum architectures based on the XZZX surface code yield remarkably high memory thresholds and low overhead as compared with the conventional surface code approach. Our generalised fault-tolerant decoder can realise these advantages over a broad range of biased error models representing what is observed in experiments for a variety of physical qubits.

The performance of the XZZX code is underpinned by its exceptional code-capacity thresholds, which match the performance of random coding (hashing) theory, suggesting that this code may be approaching the limits of what is possible. In contrast to this expectation, the XZZX surface code threshold is numerically observed to exceed this hashing bound for certain error models, opening the enticing possibility that random coding is not the limit for practical thresholds. We note that for both code capacities and fault-tolerant quantum computing, the highest achievable error thresholds are not yet known.

We emphasise that the full potential of our results lies not just in the demonstrated advantages of using this particular architecture, but rather the indication that further innovations in codes and architectures may still yield significant gains in thresholds and overheads. We have shown that substantial gains on thresholds can be found when the code and decoder are tailored to the relevant noise model. While the standard approach to decoding the surface code considers Pauli-X and Pauli-Z errors

separately, we have shown that a tailored non-CSS code and decoder can outperform this strategy for essentially all structured error models. There is a clear avenue to generalise our methods and results to the practical setting involving correlated errors arising from more realistic noise models as we perform fault-tolerant logic. We suggest that the theory of symmetries[10,37] may offer a formalism to make progress in this direction.

Because our decoder is based on minimum-weight matching, there are no fundamental obstacles to adapt it to the more complex setting of circuit noise[47,56,65]. We expect that the high numerical thresholds we observe for phenomenological noise will, when adapted to circuit level noise, continue to outperform the conventional surface code, especially when using gates that preserve the structure of the noise[27,28]. We expect that the largest performance gains will be obtained by using information from a fully characterised Pauli noise model[66–68] that goes beyond the single-qubit error models considered here.

Along with high thresholds, the XZZX surface code architecture can yield significant reductions in the overheads for fault-tolerant quantum computing, through improvements to the sub-threshold scaling of logical error rates. It is in this direction that further research into tailored codes and decoders may provide the most significant advances, bringing down the astronomical numbers of physical qubits needed for fault-tolerant quantum computing. A key future direction of research would be to carry these improvements over to codes and architectures that promise improved (even constant) overheads[39,40,42]. Recent research on fault-tolerant quantum computing using low-density parity check (LDPC) codes that generalise concepts from the surface code[41,69–74] provide a natural starting point.

## Methods

**Optimal thresholds**. In the main text, we obtained optimal thresholds using a maximum-likelihood decoder to highlight features of the codes independent of any particular heuristic decoding algorithm. Maximum-likelihood decoding, which selects a correction from the most probable logical coset of error configurations consistent with a given syndrome, is, by definition, optimal. Exact evaluation of the coset probabilities is, in general, inefficient. An algorithm due to Bravyi, Suchara and Vargo[38] efficiently approximates maximum-likelihood decoding by mapping coset probabilities to tensor-network contractions. Contractions are approximated by reducing the size of the tensors during contraction through Schmidt decomposition and retention of only the $\chi$ largest Schmidt values. This approach, appropriately adapted, has been found to converge well with modest values of $\chi$ for a range of Pauli noise channels and surface code layouts[8,38]. A full description of the tensor network used in our simulations with the rotated CSS surface code is provided in ref. [8]; adaptation to the XZZX surface code is a straightforward redefinition of tensor element values for the uniform stabilizers.

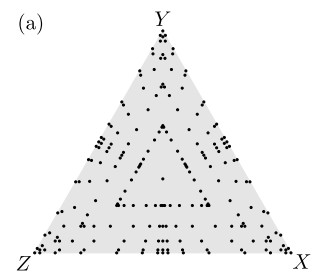
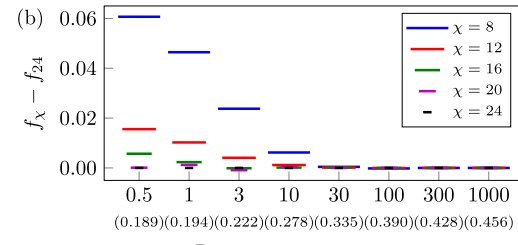

**Fig. 8 Optimal threshold sample distribution and decoder convergence. a** Distribution of 211 samples over the surface of all single-qubit Pauli noise channels. To construct each threshold surface of Fig. 2, by code symmetry, thresholds are estimated for 111 of these samples. **b** Tensor-network decoder convergence for the 77 × 77 XZZX surface code with Z-biased noise, represented by shifted logical failure rate $f_\chi - f_{24}$, as a function of truncated bond dimension $\chi$ at a physical error probability $p$ near the zero-rate hashing bound for the given bias $\eta$. Each data point corresponds to 30,000 runs with identical errors generated across all $\chi$ for a given bias.

Figure 2, which shows threshold values over all single-qubit Pauli noise channels for CSS and XZZX surface codes, is constructed as follows. Each threshold surface is formed using Delaunay triangulation of 211 threshold values. Since both CSS and XZZX square surface codes are symmetric in the exchange of Pauli-X and Z, 111 threshold values are estimated for each surface. Sample noise channels are distributed radially such that the spacing reduces quadratically towards the sides of the triangle representing all single-qubit Pauli noise channels, see Fig. 8a. Each threshold is estimated over four $d \times d$ codes with distances $d \in \{13, 17, 21, 25\}$, at least six physical error probabilities, and 30,000 simulations per code distance and physical error probability. In all simulations, a tensor-network decoder approximation parameter of $\chi = 16$ is used to achieve reasonable convergence over all sampled single-qubit Pauli noise channels for the given code sizes.

Figure 3, which investigates threshold estimates exceeding the zero-rate hashing bound for the XZZX surface code with Z-biased noise, is constructed as follows. For bias $30 \leq \eta \leq 1000$, where XZZX threshold estimates exceed the hashing bound, we run compute-intensive simulations; each threshold is estimated over four $d \times d$ codes with distances up to $d \in \{65, 69, 73, 77\}$, at least fifteen physical error probabilities, and 60,000 simulations per code distance and physical error probability. Interestingly, for the XZZX surface code with Z-biased noise, we find the tensor-network decoder converges extremely well, as summarised in Fig. 8b for code distance $d = 77$, allowing us to use $\chi = 8$. For $\eta = 30$, the shift in logical failure rate between $\chi = 8$ and the largest $\chi$ shown is less than one fifth of a standard deviation over 30,000 simulations, and for $\eta > 30$ the convergence is complete. All other threshold estimates in Fig. 3, are included for context and use the same simulation parameters as described above for Fig. 2.

All threshold error rates in this work are evaluated use the critical exponent method of ref. [45].

**The minimum-weight perfect-matching decoder.** Decoders based on the minimum-weight perfect-matching algorithm[43,44] are ubiquitous in the quantum error-correction literature[5,14,37,45,75]. The minimum-weight perfect-matching algorithm takes a graph with weighted edges and returns a perfect matching using the edges of the input graph such that the sum of the weights of the edges is minimal. We can use this algorithm for decoding by preparing a complete graph as an input such that the edges returned in the output matching correspond to pairs of defects that should be locally paired by the correction. To achieve this we assign each defect a corresponding vertex in the input graph and we assign the edges weights such that the proposed correction corresponds to an error that occurred with high probability.

The runtime of the minimum-weight perfect-matching algorithm can scale like $O(V^3)$ where $V$ is the number of vertices of the input graph[44], and the typical number of vertices is $V = O(pd^2)$ for the case where measurements always give the correct outcomes and $V = O(pd^3)$ for the case where measurements are unreliable.

The success of the decoder depends on how we choose to weight the edges of the input graph. Here we discuss how we assign weights to the edges of the graph. It is convenient to define an alternative coordinate system that follows the symmetries of the code. Denote by $f \in \mathcal{D}_j$ sets of faces aligned along a diagonal line such that $S = \prod_{f \in \mathcal{D}_j} S_f$ is a symmetry of the code with respect to Pauli-Z errors, i.e. $S$ commutes with Pauli-Z errors. One such diagonal is shown in Fig. 1(e). Let also $\mathcal{D}_j'$ be the diagonal sets of faces that respect symmetries introduced by Pauli-X errors.

Let us first consider the decoder at infinite bias. We find that we can decode the lattice as a series of one-dimensional matching problems along the diagonals $\mathcal{D}_j$ at infinite bias. Any error drawn from the set of Pauli-Z errors $\mathcal{E}^Z$ must create an even number of defects along diagonals $\mathcal{D}_j$. Indeed, $S = \prod_{f \in \mathcal{D}_j} S_f$ is a symmetry with respect to $\mathcal{E}^Z$ since operators $S$ commute with errors $\mathcal{E}^Z$. In fact, this special case of matching along a one-dimensional line is equivalent to decoding the repetition code using a majority vote rule. As an aside, it is worth mentioning that the

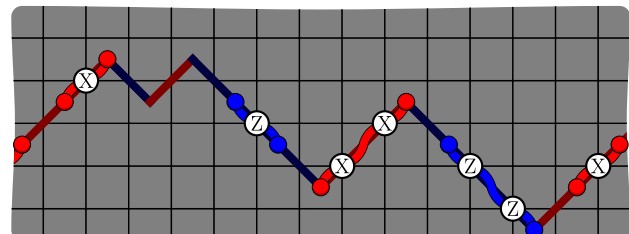

**Fig. 9 A low-weight error that causes a logical failure.** The error consists of ~$d/4$ high-rate errors and ~$d/4$ low-rate errors along the support of a weight $d$ logical operator.

parallelised decoding procedure we have described vastly improves the speed of decoding in this infinite bias limit.

We next consider a finite-bias error model where qubits experience errors with probability $p$. Pauli-Z errors occur at a higher rate, $p_{\text{h.r.}} = p\eta/(\eta + 1)$, and Pauli-X and Pauli-Y errors both occur at the same low error rate $p_{\text{l.r.}} = p/2(\eta + 1)$. At finite bias, string-like errors can now extend in all directions along the two-dimensional lattice. Again, we use minimum-weight perfect matching to find a correction by pairing nearby defects with the string operators that correspond to errors that are likely to have created the defect pair.

We decode by giving a complete graph to the minimum-weight perfect-matching algorithm where each pair of defects $u$ and $v$ are connected by an edge of weight $\sim -\log \text{prob}(E_{u,v})$, where $\text{prob}(E_{u,v})$ is the probability that the most probable string $E_{u,v}$ created defects $u$ and $v$. It remains to evaluate $-\log \text{prob}(E_{u,v})$.

For the uncorrelated noise models we consider, $-\log \text{prob}(E_{u,v})$ depends, anisotropically, on the separation of $u$ and $v$. We define orthogonal axes $x'(y')$ that align along (run orthogonal to) the diagonal line that follows the faces of $\mathcal{D}_j$. We can then define separation between $u$ and $v$ along axes $x'$ and $y'$ using the Manhattan distance with integers $l_{x'}$ and $l_{y'}$, respectively. On large lattices then, we choose $-\log \text{prob}(E_{u,v}) \propto w_{\text{h.r.}} l_{x'} + w_{\text{l.r.}} l_{y'}$ where

$$w_{\text{l.r.}} = -\log\left(\frac{p_{\text{l.r.}}}{1-p}\right), \quad w_{\text{h.r.}} = -\log\left(\frac{p_{\text{h.r.}}}{1-p}\right). \quad (5)$$

The edges returned from the minimum-weight perfect-matching algorithm[43,44] indicate which pairs of defects should be paired. We note that, for small, rectangular lattices with periodic boundary conditions, it may be that the most probable string $E_{u,v}$ is caused by a large number of high-rate errors that create a string that wraps around the torus. It is important that our decoder checks for such strings to achieve the logical failure rate scaling like $O(p_{\text{h.r.}}^{d^2/2})$. We circumvent the computation of the weight between two defects in every simulation by creating a look-up table from which the required weights can be efficiently retrieved. Moreover, we minimise memory usage by taking advantage of the translational invariance of the lattice.

We finally remark that our minimum-weight perfect-matching decoder naturally extends to the fault-tolerant regime. We obtain this generalisation by assigning weights to edges connecting pairs of defects in the $2 + 1$-dimensional syndrome history such that

$$-\log \text{prob}(E_{u,v}) \propto l_{x'} w_{\text{h.r.}} + l_{y'} w_{\text{l.r.}} + l_t w_t, \quad (6)$$

where now we have $l_t$ the separation of $u$ and $v$ along the time axis, $w_t = -\log\left(\frac{q}{1-q}\right)$ and $q = p_{\text{h.r.}} + p_{\text{l.r.}}$. In the limit that $\eta = 1/2$ our decoder is equivalent

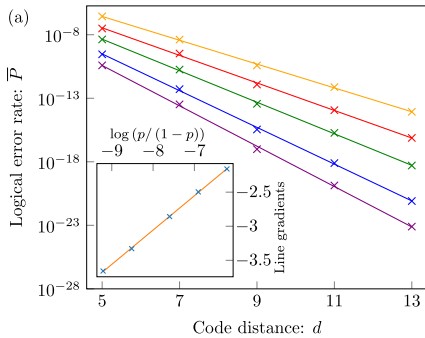
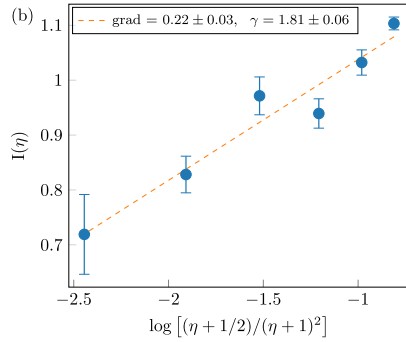

**Fig. 10 Analysis of the XZZX code at low error rates. a** Plot showing logical failure rate $\overline{P}$ as a function of code distance $d$ for data where noise bias $\eta = 3$. The physical error rates used are, from the bottom to the top curves in the main plot, 0.0001, 0.0002, 0.0005, 0.001 and 0.002. We estimate $G(p, \eta)$ for each physical error rate $p$ by taking the gradient of each line. Low error rate data are collected using the method proposed in refs. [76,77]. The inset plot shows the gradients $G(p, \eta)$ as function of $\log[p/(1-p)]$ for $\eta = 3$. Values for $G(p, \eta)$, see Eq. (8), are estimated using the linear fittings. Gradient of line of best fit to these data points is 0.504(4) in agreement with the expected gradient 1/2. **b** Plot showing intercepts $I(\eta)$ shown as a function of $\log[(\eta + 1/2)/(\eta + 1)^2]$. The intercept function is defined in Eq. (9) and estimated from the intercept of lines such as that shown in the inset of plot **a**. Error bars indicate one standard deviation relative to the fitting procedure.

to the conventional minimum-weight perfect-matching decoder for phenomenological noise[45].

**Ansatz at low error rates**. In the main text we proposed a regime at low error rates where the most common cause of logical failure is a sequence of $\sim d/4$ low-rate and $\sim d/4$ high-rate errors along the support of a weight $d$ logical operator; see Fig. 9. Here we compare our ansatz, Eq. (3) with numerical data to check its validity and to estimate the free parameter $\gamma$.

We take the logarithm of Eq. (3) to obtain

$$\log \overline{P}_{\text{lin.}} \sim n\log(1-p) + \gamma d\log 2 + \frac{d}{2}\log\left[\frac{p}{1-p}\right] + \frac{d}{4}\log\left[\frac{\eta + 1/2}{(\eta + 1)^2}\right]. \quad (7)$$

Neglecting the small term $n\log(1-p)$ we can express the this equation as $\log \overline{P} \approx G(p, \eta)d$ where we have the gradient

$$G(p, \eta) = \frac{1}{2}\log\left[\frac{p}{1-p}\right] + \gamma\log 2 + \frac{1}{4}\log\left[\frac{\eta + 1/2}{(\eta + 1)^2}\right]. \quad (8)$$

In Fig. 10a we plot the data shown in the main text in Fig. 6b as a function of $d$ to read the gradient $G(p, \eta)$ from the graph. We then plot $G(p, \eta)$ as a function of $\beta = \log[p/(1-p)]$ in the inset of Fig. 10a. The plot reveals a gradient $\sim 0.5$, consistent with our ansatz where we expect a gradient of 1/2. Furthermore, at $p = 0$ we define the restricted function

$$I(\eta) \equiv G(p = 0, \eta) = \gamma\log 2 + \frac{1}{4}\log\left[\frac{\eta + 1/2}{(\eta + 1)^2}\right]. \quad (9)$$

We estimate $I(\eta)$ from the extrapolated $p = 0$ intercepts of our plots, such as shown in the inset of Fig. 10a, and present these intercepts a function of $\log[(\eta + 1/2)/(\eta + 1)^2]$; see Fig. 10b. We find a line of best fit with gradient $0.22 \pm 0.03$, which agrees with the expected value of 1/4. Moreover, from the intercept of this fit, we estimate $\gamma = 1.8 \pm 0.06$, which is consistent with $3/2 \le \gamma \le 2$ that we expect[49]. Thus, our data are consistent with our ansatz, that typical error configurations lead to logical failure with $\sim d/4$ low-rate errors.

## Data availability
The data that support the findings of this study are available at https://bitbucket.org/qecsim/qsdxzzx/.

## Code availability
Software for all simulations performed for this study is available at https://bitbucket.org/qecsim/qsdxzzx/ and released under the OSI-approved BSD 3-Clause licence. This software extends and uses services provided by qecsim[78,79], a quantum error-correction simulation package, which leverages several scientific software packages[44,80–82].

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

## Acknowledgements

We are grateful to A. Darmawan, A. Grimsmo and S. Puri for discussions, to E. Campbell and C. Jones for insightful questions and comments on an earlier draft, and especially to J. Wootton for recommending consideration of the XZZX code for biased noise. We also thank A. Kubica and B. Terhal for discussions on the hashing bound. This work is supported by the Australian Research Council via the Centre of Excellence in Engineered Quantum Systems (EQUS) project number CE170100009, and by the ARO under Grant Number: W911NF-21-1-0007. B.J.B. also received support from the University of Sydney Fellowship Programme. Access to high-performance computing resources was provided by the National Computational Infrastructure (NCI Australia), an NCRIS enabled capability supported by the Australian Government, and the Sydney Informatics Hub, a Core Research Facility of the University of Sydney.

## Author contributions

J.P.B.A. produced the code and collected the data for simulations with the minimum-weight perfect-matching decoder, and D.K.T. wrote the code and collected data for simulations using the maximum-likelihood decoder. All authors, J.P.B.A., D.K.T., S.D.B., S.T.F. and B.J.B., contributed to the design of the methodology and the data analysis. All authors contributed to the writing of the manuscript.

## Competing interests

The authors declare no competing interests.
