## [Peer Review File · Nature Communications]

Reviewers' Comments:

Reviewer #1:

Remarks to the Author:

This is an interesting paper which presents a class of codes which outperforms the surface code in particular when the noise is biased. This kind of improvement in code performance is important with the possible arrival of noise-bias preserving gates, it also stimulates the development of such gates. I think this paper merits publication in Nature Comm, but I also believe that the authors should clarify and sharpen several aspects of this paper before it can be published.

Hashing Bound: the authors make many references to the hashing bound ("surprising effect, nonadditivity of coherent information") and what it means that the numerical data go beyond it, without stating what this bound says or what its proof is (although a protocol can be found on Wikipedia).

I believe the authors might have a misunderstanding concerning the "applicability of this bound". The authors consider the encoding of a single qubit into d^2 qubits and at what depolarising probability p it is possible to retrieve the qubit after passing through the channel with logical error going to 0 as a function of d .

However in order for a channel to have any capacity, we have to send k qubits through n channels such that $k/n > 0$ and the logical error goes to zero. The hashing bound shows that this rate k/n can be $1-H(p)$ by an explicit protocol using random stabiliser codes where p is the depol. error rate.

For the XZZX code when we encode k logical qubits, we use $n=d^2 k$ qubits and in order for the logical error rate to vanish, d should become large, so the rate k/n goes to zero. Thus the fact that the XZZX code can handle depol noise beyond p set by $1-H(p)=0$, does not seem to have a relation to "nonadditivity of coherent information" or hashing bounds. This code is not a way to achieve some finite capacity at all (maybe generalisations of the code to finite rate homological codes such as hyperbolic surface codes are, but the XZZX version of these codes is not immediately obvious).

Of course the authors are free to compare their data with the hashing bound, but the implication of 'going beyond this' are not clear besides just showing that the code performance is quite good. In fact, is it correct that there are no other LDPC stabiliser codes which go beyond this hashing bound?

line 266 "we weight each edge" -> we weigh each edge. Same sentence: if the probability is zero, then weight is $-\infty$ according to your definition, so if you want to minimise weight, you match, which does not seem to be what you want (it is also different from Eq. (B1))

Sentence "We regard a given sample...code deformation" is not clear. An error string, a mix of qubit and measurement errors occurs, the decoder determines a matching which gives the inference where qubit and where measurement errors happened. Then one checks whether whether all qubits errors, those which actually occurred and the inferred correction give rise to a logical error. I am not sure why it is necessary to say 'one has to check for temporal errors', errors are spatial-temporal pieces of strings. I presume the last round of QEC is error free as is commonly done?

The authors write in line 287 that they simulate the XZZX code with periodic boundaries. To me it is not clear that this code has the same performance as the code with open boundaries. For example, in the case of infinite noise bias they write that one decodes along diagonals "independent repetition codes" which become separate diagonal sheets in 3D, but if we have periodic boundaries in space, don't these diagonals merge and wrap around (what am I missing

here)? Did the authors consider or have data for open boundaries or can argue that they expect the performance to be similar (not based on similarity in performance for the surface code versus the toric code).

Line 357. Why are the prefactors $N_{h,r}$ and $N_{l,r}$ exponential rather than polynomial in d and d^2 .

Section VI. It is nice to show that lattice code surgery can go through with this code. However, given that the stabiliser subspace of the code is some Hadamard rotations away from that of the regular surface code (when qubits are on edges, one does Hadamards on all horizontal edges to go from the Levin-Wen model to Kitaev's toric code), can one not just take all lattice code surgery moves for the surface code and add on these Hadamards in the parity check circuits and one is done. The fact that the XZZX code is a bunch of Hadamards away from the regular surface code may be useful to mention anyhow.

In Appendix B the authors explain the details regarding their implementation of the MWPM decoder for the XZZX code. There the authors initially consider the case at infinite bias, in this case towards Z-type errors. For such a noise model Z-type errors lead to defects along the diagonal of the code, which the authors also identify as the symmetry of the code. However, when considering the case where the noise exhibits a finite bias (such that is a finite probability of X-type and Y-type errors occurring) this symmetry is broken and matching is allowed along either diagonal of the code. On one hand this should mean that one has to perform the minimum-weight perfect matching using the full syndrome volume. This can be compared to the case of the CSS surface code, where the MWPM decoding can be done separately for the X-type and Z-type syndromes. It would be nice to have a comment on how this would affect the decoder runtime. More importantly, the two diagonals that are considered for the pairing of the defects correspond to either Z-type or X-type errors. It would be nice to make more explicit that for this decoder, the correlations between errors resulting from Pauli-Y errors are also not taken into account and that this leads to the suboptimal performance of the decoder. This is especially because in section II they mentioned that pauli-Y errors can be decoded as done in a previous work from the authors (in which case they consider decoding along a different symmetry), but it isn't made clear that these symmetries are mutually exclusive.

Reviewer #2:

None

Reviewer #3:

Remarks to the Author:

Referee Report - The XZZX surface code

Summary

.....

In this work the authors present results to show that the XZZX surface code achieves impressive performance when subject to biased noise. For uniform depolarising noise, the results show that the performance of the XZZX code matches that of the standard CSS surface code. For all other biased noise channels, the code significantly outperforms previous decoding numerics for the surface code. Surprisingly, in the high-bias regime, the authors claim the XZZX code achieves performance that exceeds the Hashing bound.

The paper contains numerical decoding results for the ZXXZ code for both the code capacity case and the phenomenological noise model with noisy syndrome readout. In both settings, the authors find improved performance over the standard CSS surface code. There is also a section that explains how the advantages of the ZXXZ surface code can be maintained when performing fault

tolerant quantum computation on the encoded states.

The XZZX surface code is a variant of the standard CSS surface code and was first proposed by Xiao-Gang Wen in 2002. To our knowledge, the paper we are reviewing today is the first practical numerical study of the ZXXZ code in the context of quantum computing.

In our opinion this paper contains novel results of timely and topical interest to the quantum computing research community. Quantum computing experiments have now matured to the extent at which implementing error correction is a practical possibility. Error correction protocols that can be tailored to the specifics of a device's error model, such as the one presented in this paper, will play an important role in helping current and near-future quantum devices reach their full potential. As such, we recommend that this paper is accepted for publication in Nature Communications.

Below, we make various comments to the authors that we believe will improve the readability of the paper. We recommend that they make these changes before submitting a final manuscript for publication.

General Comments to the authors

.....

Section II

Our primary concern is that the introductory material is not sufficiently self-contained for the reader to properly understand the construction of the XZZX code. We recommend that you expand this section, or write a more detailed appendix with explicit examples of the code construction. As a minimum, the following questions should be addressed:

- 1) What are the code parameters? Does this code encode a single logical qubit as is the case with the CSS version of the surface code?
- 2) How are the logical operators defined? In Section V, you claim the Pauli-Z distance is $d(d+1)$ for a $d \times (d+1)$ lattice. However, it is not clear how these logical operators arise. What about the distance for X and Y errors? These questions should be addressed in Section II.
- 3) The XZZX code on a $d \times d$ lattice and on a $d \times (d+1)$ lattice have different distances. For lattices of sizes $m \times n$, m and n coprime, the code has distance mn for one of the errors. Your simulations are run on different lattice sizes and this influences the results (see lines 289 and 397). Why was it not possible to run the simulations for both the fault tolerant threshold and sub-threshold scaling on the same lattice. Could you explain this decision in the text?

Perhaps the best way of illustrating how the logical operators are defined is to draw the smallest example of a XZZX code, and directly show how the logical operators and stabilisers arise. In addition, it would be useful if you could release your code for generating the ZXXZ code parity check matrices. This would help interested readers build the codes and verify your results.

Section II - The Hashing Bound

- 1) The Hashing bound should be defined at least qualitatively in the introductory material. A more detailed definition should also be included in Section III.
- 2) You claim that the results could potentially provide evidence of the superadditivity of coherent quantum information. Could you expand upon this point?

3) In the limit of infinite bias, the X- and Z- decoding problems amount to classical decoding of repetition codes. Could it be that the Hashing Bound is exceeded in this regime because the error channel under consideration is a classical channel rather than a quantum one? Could a better bound be the classical Shannon Capacity, as this would not be exceeded by repetition code decoding?

Section IV

Line 325 - "It follows that by extending the syndrome along the temporal direction to account for the phenomenological noise model with infinite bias, we effectively decode decoupled copies of the two-dimensional surface code. With the minimum-weight perfect matching decoder we therefore expect a fault-tolerant threshold $\sim 10.3\%$ [14]."

-- It took several passes to understand exactly how the 2D surface code arises when you decode along the temporal axis. We think this could be explained more clearly, perhaps using a figure.

Minor comments to the authors

.....

Line 167 - "It follows that, for a noise model described by independent Pauli-Z errors, this code has a threshold error rate of 50%."

-- Are we again in the rectangular lattice settings where $d=N$ and therefore the code has correction capability $d/2$?

Line 317.

-- What are hook errors?

Fig 5.

-- Is it possible to have, in addition, a complete plot i.e. Logical failure rate at high bias / Logical failure rate at modest bias?

What about including the same plot for both cases: i.e. distance-logical failure rate and physical error rate - logical failure rate?

Line 740.

-- E_s , should it be $E_{\{u, v\}}$?

Best Wishes

Joschka Roffe & Armanda Ottaviano Quintavalle
University of Sheffield, UK

Sydney Nanoscience Hub,
University of Sydney,
NSW 2006,
Australia

February 4, 2021

Dear Referees,

Many thanks for your positive reviews, and for your constructive comments. They have certainly helped us to improve the presentation of our work. We hope you agree that with the changes we have made that our work is suitable for publication in Nature Communications.

In response to Reviewer #1

Many thanks for your detailed, constructive comments. We have made the following changes accordingly.

- “The implications of ‘going beyond [the hashing bound]’ are not clear besides just showing that the code performance is quite good. In fact, is it correct that there are no other LDPC stabilizer codes which go beyond this hashing bound.”

Reviewer #3 also asks to expand on this point.

RESPONSE: Indeed, we know of no quantum LDPC codes that go beyond the hashing bound. As such, the magnitude of the thresholds we have obtained are surprisingly high.

In regards to the nonadditivity of coherent information - In fact, such a result implies that one can produce a code that can send quantum information at finite rate where physical qubits experience an error rate above the hashing bound, even if the code has a vanishing rate. This can be achieved by concatenation. Let’s assume we have a code with $R = k/n > 0$ that has a finite threshold. Such codes are demonstrated, for instance, in Ref. 66 or 69. Then we can concatenate this code with constant sized XZZX code qubits constructed with physical qubits with error rate $p_{\text{hashing}} < p < p_{\text{threshold}}$. Let’s assume that each XZZX code requires a constant N qubits to find a logical error rate below the threshold of a finite rate code, then we obtain a new code with finite rate $R' = k/nN > 0$ for constant N using qubits with $p > p_{\text{hashing}}$.

We have added this discussion in the paragraph starting on line 242.

- Typos “We weight each edge” and “if the probability is zero then the weight is $-\infty$...”

RESPONSE: Thanks for identifying these two points. We have fixed these two sentences.

- “I am not sure why it is necessary to say ‘one has to check for temporal errors’.”

RESPONSE: In a simulation as you have described, temporal logical errors have no effect. However, in general, temporal logical errors can affect the logical state of a code deformation. As such, a threshold calculation should account for this possibility. This is discussed in Vuillot et al. 2019. We have added this reference at the end of the following sentence explaining the need to check for temporal errors.

“It is important to check for temporal errors, as they can cause logical errors when we perform fault-tolerant logic gates by code deformation [46].”

See line 317.

To make it possible to introduce a temporal logical error, we make our simulation periodic in the time direction. This approach is also common in the literature; it was adopted in early works by Raussendorf and coworkers on topological fault-tolerant quantum computation, for instance. Of course, as the system size diverges, the choice of boundary conditions should be negligible on the threshold error rates.

- “The authors write that they simulate the XZZX code with periodic boundary conditions. To me it is not clear that this code has the same performance as the code with open boundaries.”

Here we also respond to the point of reviewer #3 who asks:

“Why was it not possible to run the simulations for both the fault tolerant threshold and sub-threshold scaling on the same lattice. Could you explain this decision in the text?”

RESPONSE: We thank the referees for identifying this point. Firstly, we note that we do not expect the threshold will be affected very much by the choice of boundary conditions, because the threshold is defined in the thermodynamic limit.

The choice of boundary conditions can however have a significant affect on the logical failure rates below threshold, and, specifically, the mechanisms that cause logical failures. This was noticed in Ref. [8] with infinite biased codes with coprime lattice dimension where the effective distance of the code against the infinitely biased noise model was much higher than the actual distance of the code. Our goal here was to probe what such properties mean at finite noise bias. The XZZX code has permitted us to study this physics using our high-speed, practical MWPM decoder as a toy model to evaluate its behaviour. To do so, we chose the XZZX code with periodic boundary conditions with lattice dimensionality of $d \times d + 1$ to obtain a code with these properties. We have motivated this study over the first three paragraphs of the sub-threshold scaling subsection, and conclude by finding that in general, \bar{P}_{lin} is the dominant scaling.

On reflection, our introductory paragraphs on the sub threshold scaling section were not well written, and this has clearly led to confusion. We have updated the introduction to this section to make it clearer why we are making this choice. Please see the new motivating paragraphs starting at line 382.

- “Line 357 - Why are the prefactors exponential?”

RESPONSE: This is a well-studied term when evaluating logical failure rates. See for instance Dennis et al., or Beverland et al. Ref. [49]. It counts the ways that $\sim w/2$ errors can be configured over the support of a weight w logical operator. Using Stirling’s approximation we obtain that (w choose $w/2$) $\sim 2^w$. Indeed, while one can include a polynomial term in $N_{\text{l.r.}}$, this is subleading, and negligible compared to the exponential terms. We have therefore omitted it. To help make this a little more explicit, we have written

“where $N_{\text{h.r.}} \sim 2^{d^2}$ is the number of configurations that $d^2/2$ Pauli-Z errors can take on the support of the weight- d^2 logical operator to cause a failure.”

below Eqn. (2). We also give reference to Beverland et al. to explain the exponential term for $N_{\text{l.r.}}$.

- “It is nice to show that lattice surgery can go through with this code.” and “. . . The fact that the XZZX code is a bunch of Hadamards away from the regular surface code may be useful to mention anyhow.”

RESPONSE: Thanks for this comment. Despite the simple relationship using local rotations, it was not obvious that the high thresholds that come from structured noise can carry through to quantum computation.

- “In Appendix B: (1) It would be nice to have a comment on how this would affect the decoder runtime. (2) That correlations between errors resulting from Pauli-Y errors are also not taken into account. (3) Especially given we mentioned that Pauli-Y errors can be decoded as done in previous work.”

RESPONSE: (1) Indeed, the decoding runtime is $O(V^3)$ in the general case where V is the number of vertices for the input graph for minimum-weight perfect matching, and $V = O(pd^2)$ for the case with ideal stabilizer measurements and $V = O(pd^3)$ for the phenomenological noise model. In the first paragraph of this methods section we have written

“The runtime of the minimum-weight perfect matching algorithm can scale like $O(V^3)$ where V is the number of vertices of the input graph [44], and the typical number of vertices is $V = O(pd^2)$ for the case where measurements always give the correct outcomes and $V = O(pd^3)$ for the case where measurements are unreliable.”

(2) We have explicitly added that the minimum-weight perfect-matching decoder is suboptimal. See line 290.

(3) Finally, we also made it explicit on page 2 that we are only proposing to use the decoder from our previous paper to decode Pauli-Y errors in the limit that the noise model is highly biased towards Pauli-Y errors. We have amended the statement to read:

“We therefore see that the high-performance decoders presented in Refs. [7,8,10] are readily adapted for the XZZX code in the limit that error model is highly likely to introduce Pauli-Y errors.”

In response to Reviewer # 3

We are grateful for your suggestions. We have made the following changes in response.

Primary comments

- “What are the code parameters? Does this code encode a single logical qubit as is the case with the CSS version of the surface code?”

RESPONSE: We have elaborated on the description of the XZZX code to include the code parameters; line 122.

“They differ by a Hadamard rotation on alternate qubits Ref. [33,34]. The code parameters of the surface code are invariant under this rotation. The XZZX code therefore encodes $k = O(1)$ logical qubits using $n = O(d^2)$ physical qubits where the code distance is d . Constant factors in these values are determined by details such as the orientation of the square-lattice geometry and boundary conditions. See Fig. 1 and its caption for a description.”

We address the following two referee comments together:

- “How are the logical operators defined? In Section V you claim the Pauli-Z distance is $d(d+1)$ for a $d \times (d+1)$ lattice. However it is not clear how these logical operators arise. What about the distance for X and Y errors? These questions should be addressed in Section II”

and

- “Perhaps the best way of illustrating how the logical operators are defined is to draw the smallest

example of a XZZX code”

RESPONSE: We have added an illustration of a logical operator to Figure 1. Our preference is to show a logical operator on a larger lattice so the reader can see how the distance scales with system size. As we have already addressed in the first point, the code distance for general Pauli operators is $O(d)$.

We have also offered some intuition for how the high-weight Pauli-Z logical operators arise on the $d \times (d + 1)$ periodic lattice on line 398.

“Note we can regard this single logical-Z operator as a string that coils around the torus many times such that it is supported on all n qubits.”

- “Why was it not possible to run the simulations for both the fault tolerant threshold and sub-threshold scaling on the same lattice. Could you explain this decision in the text?”

RESPONSE: Reviewer #1 has asked a question relating to this point. We have addressed it above together with the comment of Reviewer #1; please see our response above. In brief: It is possible to run both of these simulations for both lattices. However, in both situations we study different effects.

- “In addition, it would be useful if you could release your code for generating the ZXXZ code parity check matrices. This would help interested readers build the codes and verify your results.”

RESPONSE: We have made our code and data available in a public repository under an OSI-approved licence and referenced the repository in the Data and Code Availability sections of the paper.

Section II

- “The hashing bound should be defined at least qualitatively in the introductory material. A more detailed definition should be included in section III.”

RESPONSE: We agree. We have added a qualitative definition for the hashing bound in the caption of Fig. 2. In addition, in the introduction we have written on line 63:

“... , matching what is known to be achievable with random coding according to the hashing bound.”

This identifies the hashing bound qualitatively as the bound obtained by studying the rate that information can be sent by random codes.

- “You claim that the results could potentially provide evidence of the super additivity of coherent quantum information. Could you expand on this point?”

RESPONSE: Indeed, we have given this point some more consideration since our first submission. A similar question was also raised by Reviewer #1. We have addressed these together which meant expanding on this point. Please see our response to Reviewer #1 on this point.

- “In the limit of infinite bias, the X- and Z- decoding problems amount to classical decoding of repetition codes. Could it be that the hashing bound is exceeded in this regime because the error channel under consideration is classical rather than a quantum one? Could a better bound by the classical Shannon capacity?”

RESPONSE: Thanks for this comment. We think these questions are beyond the scope of the paper, nevertheless, we can be a little more speculative in our response here. We don't think this is a purely classical phenomenon. Classically, the hashing bound is an equality and, in fact, we don't exceed the 50 percent threshold in the infinite bias (classical) limit. We only beat hashing at high, finite bias as we approach the infinite bias limit. In the quantum setting codes have degeneracy that mean the quantum version of the hashing argument is a lower bound. Other examples before our work have been found that marginally exceed the hashing bound, see Refs. [17-23] in our manuscript. The only upper bound we have on threshold is the no-cloning bound that limits the threshold against depolarising noise to $p=25$ percent. See [PRA 54, 3824 (1996)] and [JMP 49, 102014 (2008); page 14 theorem 7].

Section IV

- “It took us several passes to understand exactly how the 2D surface code arises when you decode along the temporal axis.”

RESPONSE: We have added a figure on the left hand side of what is now Fig.4. Its explanation is given in the caption.

Minor comments

- “this code has a threshold error rate of 50 percent [for independent Pauli-Z errors]? - Are we again in the rectangular lattice setting?”

RESPONSE: This is a generic statement that is true independent of the boundary conditions.

- “What are hook errors?”

RESPONSE: Hook errors are correlated errors that result from errors that occur midway through a stabilizer readout circuit. We do not model them with our phenomenological simulation. They appear when stabilizer readout circuits are modelled. We have given reference to the original work where they were seriously considered in decoding algorithms; these are Fowler et al. 2011 (Ref. [47]) as well as a comprehensive study of circuit noise by Ashley Stephens (Ref. [48]). We have also slightly reworded the sentence (line 341) to give a passing definition of a hook error. It now reads:

“We also remark that hook errors, i.e., correlated errors [47,48] that are introduced by this readout circuit, are low-rate events as the control qubit of entangling gates commute with the high-rate Pauli-Z errors, and so high-rate errors are not spread to the code.”

- “Is it possible to have, in addition, a complete plot i.e., logical failure rate at high bias/ logical failure rate at modest bias?”

RESPONSE: The data we have presented reflects the physics of the whole system, and the graphs we have presented accentuates the behaviour of the different scaling regimes we have predicted. Let us explain our choice to present the plots we have. We have proposed two regimes; that where $\overline{P}_{\text{lin}}$ dominates, and that where $\overline{P}_{\text{quad}}$ dominates. The behaviour of the system moving over this transition is very difficult to model in general. However, by collecting data in some more extremal limits of the phase space where numerics are tractable, our numerics show that our hypothesis is correct, as our numerics agree with the model we propose for our data. The intermediate regimes simply interpolate between the quadratic and linear scaling regime. In general, when either p vanishes, or when system size diverges, $\overline{P}_{\text{lin}}$ dominates. It is in these

regimes where we expect to operate a large scale quantum computer. In this sense we really have modelled the general behaviour of the system within system parameters that are tractable for numerical simulations. We have demonstrated our decoder performs as we expect in this regime on the right hand side of Figure 6, as we find our data agrees with our ansatz. Increasing the bias takes us out of this regime we are studying and moves us towards the quadratic scaling regime. Likewise, it is important to demonstrate the quadratic scaling regime in the context of earlier literature. This was identified in earlier work, Tuckett et al. 2019 PRX (2019), in the infinite bias limit. Indeed, the left hand side of Figure 6 demonstrates the behaviour we expect. We find the quadratic scaling diminishes as we decrease the error rate as we begin transitioning into the linear scaling regime.

- “ E_s , should it be $E_{u,v}$?”

RESPONSE: Absolutely. Thank you for spotting this mistake, which we have now corrected.

An additional change

We have collected additional thresholds for the rectangular variant of the XZZX code. Remarkably, we found that the MWPM decoder was also able to exceed the hashing bound in this simulation. We have included this data in Fig. 4.

We look forward to your response.

With kind regards,

Benjamin J. Brown, on behalf of the
authors

Reviewers' Comments:

Reviewer #1:

Remarks to the Author:

The authors have clarified several points in the paper: they have adequately addressed my previous points and I support publication.

Reviewer #2:

Remarks to the Author:

The revisions make the discussion of the hashing bound and the scaling of the logical failure rate easier to follow. The new paragraph on page 3 is especially helpful. As a point of curiosity, I wonder about the branching from the twist that is used to measure Pauli-Y in Fig 7c. Do you expect to achieve the same scaling of logical failure rate across different biases since the twist may introduce new nontrivial cycles and hook errors?

This article addresses a topic that is highly relevant given recent experimental developments and is exciting and well written. I recommend publishing it in Nature Communications.

Reviewer #3:

Remarks to the Author:

We are satisfied with the changes the authors have made to the manuscript.

In response to Reviewer #2

- “As a point of curiosity, I wonder about the branching from the twist that is used to measure Pauli-Y in Fig 7c. Do you expect to achieve the same scaling of logical failure rate across different biases since the twist may introduce new nontrivial cycles and hook errors?”

RESPONSE: In this code, twists add new nontrivial cycles that are used to encode and manipulate quantum information. In our construction, we maintain the one-dimensional symmetries of the code when we introduce twist defects, up to the specific location of the twist defect where the one-dimensional symmetries branch. We expect that we can still exploit this slightly modified structure to obtain very low logical failure rates with a relatively small resource overhead provided all of the twists remain suitably well separated.

In regards to hook errors; indeed, this is an important concern. The details of the circuits that would be used to measure the stabilizers of twists go beyond the scope of this work, as it is closely tied to the specific implementation. Nevertheless, there are works that have considered these details, see e.g. Ref.[64] where the authors look for ways of minimising the errors introduced by stabilizer readout circuits. We do not expect hook errors to affect our system any more so than any other generic implementation of the surface code with twists.

Additionally, in the examples we have considered, we have found that we can propose stabilizer readout circuits in some suitable choice of basis such that hook errors are low-rate events. Likewise, we expect that one can find readout circuits that minimise the effect of hook errors at the location of the twist stabilizer.

Of course, we are not concerned if this turns out to be an insurmountable problem as we know of methods of universal quantum computation where we do not use twist defects and, instead, we distil eigenstates of the Pauli-Y operator as a resource for S gates. In which case the circuits we have discussed in the manuscript will suffice to complete a universal set of fault-tolerant quantum computational operations

Thank you once again for your time and effort in reviewing our manuscript.